# Clinical Management of Supratentorial Non-Skull Base Meningiomas

**DOI:** 10.3390/cancers14235887

**Published:** 2022-11-29

**Authors:** Adefisayo Adekanmbi, Mark W. Youngblood, Constantine L. Karras, Ephraim A. Oyetunji, John Kalapurakal, Craig M. Horbinski, Hinda Najem, Virginia B. Hill, James P. Chandler, Amy B. Heimberger, Stephen T. Magill, Rimas V. Lukas

**Affiliations:** 1Department of Neurological Surgery, University of Wisconsin, Madison, WI 53792, USA; 2Department of Neurological Surgery, Northwestern University, Chicago, IL 60611, USA; 3Washington University, St. Louis, MO 63110, USA; 4Department of Radiation Oncology, Northwestern University, Chicago, IL 60611, USA; 5Lou & Jean Malnati Brain Tumor Institute, Northwestern University, Chicago, IL 60611, USA; 6Department of Pathology, Northwestern University, Chicago, IL 60611, USA; 7Department of Radiology, Section of Neuroradiology, Northwestern University, Chicago, IL 60611, USA; 8Department of Neurology, Northwestern University, Chicago, IL 60611, USA

**Keywords:** DOTATATE, meningioma, methylation, MRI, pathophysiology, radiation, supratentorial, surgery, systemic therapy, targeted therapy

## Abstract

**Simple Summary:**

Meningiomas are the most-common primary central nervous system tumors. These tumors are most commonly supratentorial in location. The current clinical management ranges from observation to multimodality treatment including surgery and radiation therapy. Non-invasive imaging features can help predict progression when tumor tissue is not available, in turn influencing the approach to these tumors. In parallel, there is a growing understanding regarding characteristics that influence their growth rate. This derives in large part from molecular pathologic studies that shed light on the pathophysiology of disease. It is hoped that this will impact the clinical management, particularly with regard to the use of radiation and systemic therapies.

**Abstract:**

Supratentorial non-skull base meningiomas are the most common primary central nervous system tumor subtype. An understanding of their pathophysiology, imaging characteristics, and clinical management options will prove of substantial value to the multi-disciplinary team which may be involved in their care. Extensive review of the broad literature on the topic is conducted. Narrowing the scope to meningiomas located in the supratentorial non-skull base anatomic location highlights nuances specific to this tumor subtype. Advances in our understanding of the natural history of the disease and how findings from both molecular pathology and neuroimaging have impacted our understanding are discussed. Clinical management and the rationale underlying specific approaches including observation, surgery, radiation, and investigational systemic therapies is covered in detail. Future directions for probable advances in the near and intermediate term are reviewed.

## 1. Introduction

Meningiomas are the most common primary central nervous system (CNS) tumor in adults (9.12 per 100,000 persons) and constitute 55% of non-malignant tumors in the United States [1]. Many meningiomas are likely not being captured in cancer registries and as such the true incidence is likely higher than reported [2]. Most occur in the supratentorial compartment [3,4], making supratentorial meningiomas the most common tumors of the cranial vault. Reports have dichotomized supratentorial meningiomas into non-skull base and skull base [4,5,6]. This classification is relevant as the location is associated with differences in clinical presentation, tumor grade, prognosis, and management plans. Supratentorial non-skull base meningiomas (NSBM) can be further sub-classified based on the location of dural attachment such as the convexity, falcine, or parasagittal locations [4,5]. An understanding of the unique characteristics of supratentorial NSBM will be relevant for clinicians when discussing prognosis and management with patients. This review will discuss current clinical decision-making and modern surgical and adjuvant management of supratentorial NSBM.

## 2. Epidemiology

Supratentorial NSBM account for 40 to 60% of all intracranial meningiomas [4,6,7,8,9] (Table 1). Meningioma incidence increases with age, with a median age at diagnosis of 66 years [1]. This association is most pronounced for NSBM [4,10,11,12]. As the world population ages, the observed incidence of meningiomas will likely rise. Females have more than double the age-adjusted meningioma incidence compared to males (12.76 per 100,000 vs. 5.79 per 100,000) [1], although meningiomas in males are more likely to be higher grade (World Health Organization [WHO] grade 2 and 3) [6,8] and supratentorial [3].

## 3. Pathophysiology

Meningiomas are thought to arise from the arachnoid cells, and in particular those associated with the arachnoid granulations, which comprise the arachnoid mater that covers the brain and spinal cord. This develops along a ventral-dorsal axis upon embryonic closure of the neural tube [14]. After formation of a primary meninx layer that covers the nascent CNS, progressive differentiation of meningeal layers leads to complex regional cell populations, marked by distinct gene expression profiles and biological roles [15]. The meninges are an important mediator of brain and skull development [16,17], secreting local factors that drive neuronal division, migration, and maturation [16]. Intracranially, the cellular origin of the meninges varies by location, as dorsal and posterior regions arise from mesoderm, while ventral regions (including much of the forebrain skull-base) are derived from migrating neural crest populations [18]. These distinct embryonic origins may explain clinical associations tied to meningioma location, including enrichment of specific molecular driver events and long-term behavior.

There has been a growing understanding of the role of CNS lymphatics, which reside within the meninges, since their initial description [19]. This meningeal lymphatic system demonstrates anatomic variability. This variability may be influenced by phylogeny as well as physical factors such as the high-cerebrospinal fluid pressure environment where they reside [19]. This lymphatic system plays a governing role in CNS immune surveillance [20], a role which in the future could impact therapeutic management. Immune-enriched subtypes of meningioma express extracellular matrix remodeling genes as well as meningeal lymphatic genes (*LYVE1*, *CCL21*, *CD3E*) [21]. A tailored approach may be necessary to adequately address molecularly defined meningioma subtypes. This may incorporate taking advantage of meningeal lymphatics and their role in immune regulation.

Large-cohort genomic studies have identified mutually exclusive molecular sub-groups of meningiomas, characterized by distinct driver mutations and clinical features [22,23,24,25,26,27]. The most frequent alteration is biallelic loss of the tumor suppressor *NF2*, which occurs in approximately half of all cases and is enriched in non-skull base locations. A subset of *NF2* mutant meningiomas exhibit recurrent mutations in *SMARCB1* (most frequently *SMARCB1R368H* and *SMARCB1R377H*), which, like *NF2* is located on chromosome 22q. These lesions are typically found along the midline falx, and may be associated with higher mitotic rate [24]. The remaining genomic subgroups are concentrated in neural-crest-cell-derived skull-base lesions and include activating variants in the PI3K and hedgehog signaling pathways or mutations in *TRAF7*, *POLR2A*, or *KLF4* [22,23,24,25,26,27]. Several somatic copy number events have also been associated with meningioma formation and progression [28,29,30], including loss of chromosome 22q and 1p, and to a lesser extent, chromosomes 9p and 14q. Meningiomas that exhibit multiple copy number losses (termed chromosomal instability) are more likely to be high-grade and with an elevated proliferation index [29,31].(Figure 1).

## 4. Natural History of Asymptomatic Meningioma

Incidental asymptomatic meningiomas are increasingly diagnosed during routine investigations for other clinical conditions [32]. The prevalence of incidental meningiomas in the general population is estimated at 2.5% as described in one prospective population-based brain magnetic resonance imaging (MRI) study of 5800 participants [33]. As meningiomas have variable growth rates, volumetric measurement provides a more precise estimation of tumor growth [34]. The mean annual growth rate of asymptomatic meningioma is 1cm^3^, ranging from 0.03 to 2.63 cm^3^ per year [35,36,37]. Overall, about 11% of incidentally discovered asymptomatic meningiomas will show growth progression [38,39]. This is presumably influenced by the baseline tumor volume. Symptomatic progression of an incidental meningioma is higher if the tumor demonstrates an annual volume change of 2.1cm^3^ per year [40], or if the initial tumor size is more than 4 cm in diameter [41]. The relative tumor growth rate is higher in patients younger than 60 years. Patients with large tumors may require shorter follow-ups because the rate of tumor growth appears to correspond with increased initial tumor volume [35,40]. In a clinical series analyzing 110 incidental meningiomas, 75% of supratentorial NSBMs showed growth whereas only 40% of skull base meningiomas showed growth in a 4-year time span [42]. In a systematic review of 675 untreated meningioma patients, 11% of convexity meningiomas and 13% of parasagittal/falcine meningiomas demonstrated symptomatic progression over a follow-up period of 4.6 years [43]. The European Association of Neuro-Oncology (EANO) suggests that incidental meningiomas in the elderly and asymptomatic patients can be managed with a watch and scan strategy [44]. Some authors recommend a shorter interval of follow-up neuroimaging for supratentorial NSBMs than for skull base meningiomas [42].

## 5. Clinical Presentation of Symptomatic Meningioma

The clinical presentation of symptomatic patients with supratentorial NSBMs varies depending on the tumor site, mass effect, and cerebral edema. Patients can present with either localizable symptoms (focal neurological deficit or seizures) or non-localizable symptoms (positional headaches, nausea/vomiting, diplopia, somnolence) related to increased intracranial pressure. Seizures are one of the most frequent presentations for patients with symptomatic meningiomas, affecting nearly 10–70% of supratentorial NSBM patients [44,45,46,47,48]. The presence of peritumoral edema is a strong predictor of seizures [1]. Although meningioma occurs more commonly in females and adults, a higher rate of seizures occurs in males [45,46], which may be because grade 2 and 3 meningiomas do not demonstrate the same female predominance observed with grade 1 meningiomas, [6]. Among supratentorial NSBMs, higher rates of seizures occur in patients with convexity or parasagittal meningiomas compared to falcine meningiomas [45,47]. One meta-analysis investigated the outcome of preoperative seizure control after surgery in 703 meningioma patients and reported that approximately 70% of the patients are seizure free after resection [45], although those with marked peritumoral edema have lower rates of postoperative seizure control [45,48,49]. Early surgical resection is advocated for patients with preoperative seizures to reduce the risk of developing medically refractory epilepsy [46].

Patients with convexity meningiomas typically have larger tumors at the time of symptomatic diagnosis. In these patients, cognitive and personality changes are common, occurring in ~40–50% [50,51]. Other neuroanatomic location-specific symptoms in patients with convexity meningiomas include motor, sensory, language, and executive functioning impairment. The presenting symptoms for parasagittal meningiomas depend on the relationship between the tumor and the superior sagittal sinus. Tumors adjacent to the frontal cortex may exhibit personality deficit and executive dysfunction, whereas lesions close to the parieto-occipital cortex may present with visual-spatial symptoms. Unique symptoms, such as lower limb paresis and contralateral sensory or motor seizures, suggest a peri-Rolandic parasagittal location [52]. Parafalcine meningiomas have similar clinical presentations to parasagittal meningiomas as these tumors arise at any point from anterior to posterior along the falx. Even when unilateral, they can cause symptoms localizable to the contralateral brain due to their mass effect. Interestingly, the patients with parafalcine meningiomas appear to have a reduced threshold for symptomatic presentation compared to the other supratentorial NSBMs [53]. About half of parafalcine meningiomas are located in the middle third of the falx [54], which borders the sensory and motor cortex, increasing surgical risk [55].

## 6. Imaging

Although MRI with contrast is the modality of choice for diagnosis and surgical planning, computed tomography (CT) is widely accessible for the rapid evaluation of most patients, is often the modality on which a meningioma is incidentally detected, and can be helpful if MRI is contraindicated due to non-MRI compatible implants, for example. (Figure 2 and Figure 3). On CT scan, meningiomas usually are either hyperdense or isodense to cortex. There are rare examples of hypodense meningiomas. Intralesional calcification is seen in 25% of meningiomas. The non-contrast CT scan can reveal bony changes, such as hyperostosis, remodeling or scalloping of the calvarium, as well as adjacent bone lysis or destruction. The calvarial changes may be but are not always associated with tumor invasion [56].

MRI provides relevant detail about the homogeneity/heterogeneity of the tumor and its relationship to the adjacent bone, brain and vascular structures. Meningiomas typically appear as well circumscribed extra-axial lesions on MRI. Most meningiomas are isointense to hypointense compared to gray matter on T1-weighted sequences and isointense or hyperintense on T2-weighted sequences. T2-weighted imaging best demonstrates the CSF cleft between the parenchyma and the meningioma. Displaced vessels can be noted within the CSF cleft. Intratumoral or peritumoral cysts can be seen in meningiomas, sometimes with the accumulation of proteinaceous fluid within the cysts that does not suppress on fluid attenuated inversion recovery (FLAIR) sequences. Contrast enhancement is avid and the peripheral dura may demonstrate a “dural tail” [57,58]. This, however, is not a pathognomonic finding. The dural tail usually does not but may contain tumor. Hyperostosis or osteolysis can be detected on MRI, but is best defined on CT.

Peritumoral edema is reported with meningioma in 34–60% of patients [45,59,60,61,62], although this range may be biased toward the higher side. Edema, if present, does not correlate with tumor size. Microcystic and secretory meningiomas are often associated with significant edema [63]. Additionally, more aggressive lesions, characterized by increased grade or other prognostic markers, may exhibit a greater amount of vasogenic edema [64]. Brain invasion, which can occur in grade 2 and grade 3 meningiomas, can manifest as an ill-defined, multilobulated margin along with loss of the CSF cleft sign. Joo et al. demonstrated that radiomics features at the tumor interface, increased volume of peritumoral edema, and male sex were predictive of brain invasion [65].

T2-weighted imaging can help predict intraoperative tumor texture. For example, a lesion with T2-hyperintensity is likely to be a soft tumor, whereas T2 hypo-intensity may signify a calcified or fibrous tumor [66]. Susceptibility on gradient echo T2* correlates with calcification as well.

Conventional angiography or magnetic resonance angiography (MRA) may be used for surgical planning, although this is not routine for most convexity meningiomas. Parasagittal meningiomas may be associated with superior sagittal sinus narrowing, invasion, and thrombosis. The degree of superior sagittal sinus invasion is assessed by MR or CT venography, or digital subtraction angiography. Findings on these studies may help clinicians with patient management and surgical planning. CTA, MRA, and DSA as well as contrast-enhanced cross-sectional imaging can demonstrate radiating enhancing vessels extending to peripheral margins of tumor.

Although, MRI is the current gold standard for imaging of meningiomas, advanced imaging techniques, such as somatostatin analogue positron emission tomography (PET), diffusion and perfusion imaging and MR spectroscopy hold promise to augment standard MR imaging. However, their use is not a component of routine clinical management. Diffusion weighted imaging (DWI) parameters correlate with *TERT* promoter mutation status in grade 2 meningiomas [59]. In a multivariable analysis, age and apparent diffusion coefficient (ADC) 10th percentile were predictive of *TERT* promoter mutation. In a univariable analysis, meningiomas with *TERT* promoter mutation were associated with increased diameter and increased volume [59]. Radiomic analyses have identified distinctive features of meningiomas with potential application in clinical management. Preoperative prediction of meningioma consistency as soft, intermediate, or fibrous based on the T2 intensity of tumor may help in treatment planning [67]. Radiomics features of T1-weighted contrast-enhanced imaging have been used preoperatively to determine meningioma grades [67]. Elevated ADC can preoperatively distinguish meningioma regions with proliferating cells enriched for developmental gene expression programs [68]. High ADC regions in meningiomas correlate with areas overexpressing the *CDH2* and *PTPRZ1* genes, which drive meningioma tumorigenesis and may represent novel therapeutic targets [68].

The cross-sectional imaging appearance of meningiomas can mimic other dural masses, most commonly and consequentially dural metastases. Solitary fibrous tumors of the dura or gliomas that have invaded the dura can also appear similarly. Paget’s disease and fibrous dysplasia could mimic the hyperostosis often seen with meningiomas [57,69]. Granulomatous processes and rarely extramedullary hematopoiesis, idiopathic or postsurgical hypertrophic pachymeninges, and dural hemangioma can also mimic meningiomas [56].

The use of ^18^F-fluorodeoxyglucose (FDG) correlates with the proliferative potential of meningiomas, and its uptake is increased in atypical and malignant tumors compared to lower grade meningiomas [69,70]. However, it has limited application because the high uptake by the normal brain lowers the observed tumor-to-background ratio. ^68^Ga-DOTATATE PET/MR utilizes the expression of the somatostatin receptor in meningiomas for the binding of DOTATATE a radiolabeled 8 amino acid long peptide which serves to target somatostatin receptors to better image these tumors. It is a tool with high sensitivity and accuracy for meningioma [71]. PET imaging using ^68^Ga-DOTA-based somatostatin analogs (^68^Ga-DOTATATE and DOTATOC) are the most frequently used radiotracers for somatostatin receptor (SSTR) imaging of meningiomas [72]. The high expression of SSTR in meningioma cells makes it a reliable cellular biomarker and DOTATATE specifically binds SSTR2 [73]. ^68^Ga-DOTATATE is more sensitive than contrast-enhanced MRI for delineating tumor boundaries in preparation for surgery and radiotherapy, defining residual and recurrent tumors, and determining response to therapy [73,74]. In a cohort of 64 meningioma patients, ^68^Ga-DOTATATE PET SUVmax reliably predicted future tumor growth in WHO grades 1 and 2 meningiomas [75]. The authors suggest that ^68^Ga-DOTATATE PET can assist in identifying patients with fast-growing meningioma to consider for early therapeutic intervention. In addition,^68^Ga-DOTATATE PET/CT has higher sensitivity and specificity for depicting transosseous infiltrative meningioma compared to contrast-enhanced MRI [70]. How this imaging modality will impact clinical practice is unclear. Some authors suggests that ^68^Ga-DOTATOC-PET/MRI provides the best morphological visualization of meningiomas, but given the low resolution of PET scanning, it remains to be seen how it will be incorporated in practice [76]. An important potential application of ^68^Ga-DOTATATE-PET/MRI in meningioma is that it may facilitate the development of SSTR-targeted radionuclide individualized treatment [77].

## 7. Pathology and Molecular Diagnostics

Histologically, most meningiomas consist of cells forming whorls and (eventually) mineralized psammoma bodies. Some also have intranuclear clearing and intranuclear cytoplasmic pseudoinclusions [78]. Most tumors express SSTR 2A and, to a lesser extent, epithelial membrane antigen [79]. However, there is a tremendous range of morphologic subtypes among the three WHO grades of meningioma [80]. Supratentorial NSBMs are more likely to be grade 2–3 than skull base tumors [4,6,81,82,83]. In one clinical series of 1663 patients, supratentorial NSBMs were twice as likely to be grade 2 or 3 [83]. In another series of 794 cases, they were four times more likely than skull base or spinal meningiomas to be grade 2 or 3 [84]. Similarly, supratentorial NSBMs tend to have a higher MIB-1 proliferation index [42,81,85,86]. Even when focusing just on WHO grade 1 tumors, the MIB-1 labeling index of NSBM is typically in the range of 2.6–2.7% compared to only 1.4–2.1% for skull base tumors [42,82]. Even after gross total resection (GTR), supratentorial NSBMs are more likely to recur with a higher grade than skull base meningiomas [81,82]. Conversely, meningiomas with high progesterone receptor expression are less likely to recur [87], and supratentorial NSBMs tend to have less progesterone receptor expression than skull base meningiomas [88].

*NF2*, located on chromosome 22q, is the most commonly altered gene in meningioma [24,27,89]. Deletion or other inactivating alterations of is present in 50–60% of patients with sporadic meningioma [90,91,92]. The *NF2* gene codes for merlin, a 69-kDa protein that links transmembrane receptors and signaling molecules that regulate cellular proliferation and survival [93,94]. Deletion of *NF2* activates phosphoinositide 3-kinase (PI3K), Notch, AKT, Ras/mitogen activated protein kinase, mammalian target of rapamycin (mTOR), and Hippo signaling pathways thereby enhancing cellular proliferation and tumorigenesis [89,93,95]. Overall, *NF2* alterations are present in 70–80% of grade 2–3 meningiomas and 40% of grade 1 meningiomas [90,96,97]. Likewise, supratentorial NSBMs are *NF2*-altered more often than skull base meningiomas [27,91,98,99]. Notably, even grade 1 supratentorial NSBMs with *NF2* alterations tend to be more aggressive [100]. In addition to 22q/*NF2* loss, copy number variations affecting larger chromosomal regions, including loss of chromosomes 1p, 6q, 10, 14q, and 18q, as well as gain of chromosomes 1q, 9q, 12q, 15q, 17q, and 20q, have been associated with more aggressive behavior [91,101,102,103].

Among non-NF2 meningiomas, key driver mutations involve Kruppel-like factor 4 (*KLF4*), v-Akt murine thymoma viral oncogene homolog 1 (*AKT1*), tumor necrosis factor receptor-associated factor 7 (*TRAF7*), phosphatidylinositol-4,5-bisphosphate 3-kinase catalytic subunit alpha (*PIK3CA*), smoothened (*SMO*) and RNA polymerase II subunit A (*POLR2A*) [22,27,104,105]. These genetic mutations are present in approximately 40% of sporadic meningiomas and are enriched in skull base tumors compared to the supratentorial NSBMs [91,106].

A new development in the molecular workup of meningiomas involves assessing genomic DNA methylation patterns, as differences in such patterns have been shown to be more prognostically accurate than traditional histopathologic WHO grading alone [21,107,108,109]. While such methylation-themed studies are intriguing, no classifiers have yet been adopted for routine clinical use. This is in part because basic preanalytical variables like the effect of admixed nonneoplastic cells on methylation results have not yet been established, and methylation profiling is not yet widely available. In contrast, telomerase reverse transcriptase (*TERT*) promoter mutations [110] and homozygous loss of *CDKN2A/B* [111] are associated with poor prognosis. By current WHO classification, meningiomas with these alterations are grade 3 regardless of histologic appearance [80].

## 8. Clinical Management

Once considered a homogeneously benign diagnosis, meningiomas are now more appropriately risk-stratified based on WHO grading, MIB-1 labeling index, molecular alterations, and DNA methylation profiling–these carry significant implications for clinical management. Management options include observation, surgical resection, radiation, or systemic therapies.

### 8.1. Observation

Given the increasing prevalence of incidental meningiomas, observation plays a critical role in clinical management. Generally, observation is preferred for all asymptomatic patients with incidental meningiomas [44]. According to the current National Comprehensive Cancer Network guidelines (NCCN), unresected meningiomas can be offered serial surveillance with Brain MRI at 3, 6 and 12 months after initial diagnosis, then 6–12 monthly for 5 years, then every 1 to 3 years as clinically indicated. The Incidental Meningioma: Prognostic Analysis using patient Comorbidity and MRI Tests (IMPACT) score is a system for using baseline clinical and radiographic features to categorize patients into three distinct prognostic categories. IMPACT can be used to assess risk for future growth and has an online calculator (https://www.impact-meningioma.com/) that is easily used [112,113,114,115,116]. Abnormally quick interval growth (or growth above a pre-determined threshold volume), concerning imaging features suggestive of higher grade, or younger patient age may precipitate treatment in an otherwise asymptomatic patient. Primary treatment generally consists of surgical resection, radiation, or a combination of both.

### 8.2. Surgery

Symptomatic (typically seizures or focal neurological deficits) supratentorial NSBMs due to mass effect are best addressed with surgical resection, which provides diagnostic and therapeutic benefit. Exceptions to this include cases where surgical risk outweighs potential benefits due to medical co-morbidities or patient age [45,117], although age should not be considered a contraindication [118,119]. Growing but asymptomatic lesions, particularly in younger patients or if higher risk imaging features are present, may also be surgically resected. In certain such patients, continued surveillance or primary radiosurgery are also reasonable alternatives [117].

Simpson grading, categorizes the extent of meningioma resection on a scale of 1 to 5 with lower numbers indicative of more extensive resection, and grades 1–3 all degrees of gross total resection [120]. From a technical perspective, in comparison to skull base meningiomas, a lower Simpson grade resection is typically achievable with lower morbidity in supratentorial meningiomas, except when the sagittal sinus is partially occluded [10]. However, a greater proportion of supratentorial NSBM are WHO grade 2 or 3, introducing a greater incentive to aim for a GTR [6,82]. While GTR’s are often achievable with acceptable risk in supratentorial meningiomas, perirolandic tumors and meningiomas involving the superior sagittal sinus can represent formidable challenges [121]. Surgical morbidity in these locations, particularly in higher grade tumors with pial invasion lacking an arachnoid plane, can be elevated and may warrant a subtotal resection (STR) or debulking [121].

### 8.3. Radiation

Radiation therapy (RT) is widely used for meningiomas as primary therapy, or as adjuvant or salvage therapy postoperatively. Indications depend on tumor size and location, pathology, patient age, comorbidities, and requests. The standard dose for the different radiation therapy modalities are as follows: stereotactic radiosurgery (12–16 Gy in a single fraction) [122], hypo fractionated stereotactic radiotherapy (25–30 Gy in 5 fractions) [123], and standard fractionated RT (50.4 Gy for grade 1, 54 Gy for grade 2 and 60 Gy for grade 3 meningiomas) [124,125]. These dose regimens are associated with an excellent long-term record of safety and efficacy. In addition to photon RT, particle-based RT using either protons, carbon ions or boron neutron capture, has also been evaluated, often in the reirradiation setting. Control rates are promising and toxicities generally tolerable [126,127,128]. Currently, the decision to use particle RT is made on a case-by-case basis. Further investigation is, however, required to more clearly define the role of particle RT in this disease.

Although most asymptomatic supratentorial meningiomas undergo continued surveillance, stereotactic radiosurgery (SRS) can be considered for presumed grade 1 meningiomas [129]. If imaging characteristics or growth rate are concerning for higher grade, surgery would provide both diagnostic and treatment benefits to the patient relative to the unclear role of SRS in this setting. While rare, the risk of adverse radiation events such as peritumoral edema or radiation necrosis must be weighed against the risks of continued surveillance or surgical resection [129]. Postoperatively in grade 1 meningiomas, SRS is often indicated after subtotal resection or at the time of progression/recurrence [117].

All WHO grade 3 meningiomas typically undergo fractionated radiotherapy postoperatively regardless of extent of resection [117]. GTR followed by fractionated radiotherapy is associated with increased overall survival in this cohort [130]. Systemic therapies are often utilized in this patient population as well, given the high risk of recurrence/progression, although none have been proven to be effective [130].

Active debate and true clinical equipoise are present about the role of radiation in grade 2 meningiomas. In patients undergoing STR, fractionated radiotherapy is associated with significantly improved overall survival and its benefits clearly outweigh the risk profile [130,131]. Following GTR, some studies have demonstrated improved local recurrence and progression-free survival (PFS) rates following adjuvant RT, but none have shown a significant overall survival benefit [132]. To better understand the role of RT in these patients, the NRG BN003 trial (NCT03180268) is a phase III randomized trial of observation versus fractionated RT following GTR of grade 2 meningiomas, with a primary endpoint of PFS [133]. The RTOG 0539 was another important study that evaluated the effect of intensity-modulated radiotherapy to a dose of 60 Gy in high-risk meningiomas (grade 3 or recurrent grade 2), showing a 3-year PFS of 58.8% in this high-risk cohort with very few adverse events [125].

### 8.4. Systemic Therapies

At this time, systemic therapies are typically reserved for complex, progressive/recurrent meningiomas that are not amenable to other treatment modalities (surgery or radiation). These include a mix of targeted and non-targeted therapies [44,134,135,136,137] (Table 2). Thus far, there has been no clear evidence of success. However, due to a marked unmet need, investigations of systemic therapies for this patient population persist. Using multiplatform profiling, 115 meningiomas were analyzed to prioritize drug targets including immune therapeutics [135]. Based on tumor expression, therapeutics directed toward NF2 and topoisomerase IIA would benefit the vast majority of meningioma patients. The most frequent protein target expressed was the epidermal growth factor receptor and antibody drug conjugates could be considered in this context. PD-L1 was expressed in 25% of grade 3 meningiomas, but not lower grade tumors. PD-1+ expression was present in about half of the meningiomas. Many of the current ongoing or actively planned studies are driven by a mechanistic approach addressing specific aberrancies with the tumors and supported by preclinical modeling.

A number of systemic therapies hold substantial promise. In light of the neuroanatomically mutually exclusive incidence of some specific mutations in meningiomas, NF2 is a particularly attractive target for supratentorial NSBM. NF2 targeting is being actively investigated in the ongoing Alliance A0171401 phase 2 trial (NCTO2523014). A0171401 is evaluating the Focal Adhesion Kinase (FAK) inhibitor GSK2256098, alongside other agents such as the SMO inhibitor vismodegib, the AKT inhibitor capivasertib, and the CDK4/6 inhibitor abemaciclib [136], which have less relevance in NSBM. The results for 36 patients with *NF2* mutated tumors have been reported. In grade 1 tumors (*n* = 12) the 6 month PFS was 83% and in the grade2/3 tumors (*n* = 24) 6 month PFS was 33% [136]. There is currently a trial in development by NRG Oncology which plans to utilize another CDK4/6 inhibitor ribociclib together with RT (60 Gy in 30 fractions) in high-risk meningioma (grade 3, recurrent grade 2 and sub-totally resected grade 2 tumors) to take advantage of the retinoblastoma protein mediated cell cycle progression which plays a central role in meningiomas [137]. Another promising approach, particularly for the fastest growing tumors, is the use of the microtubule inhibitor and bcl2 attenuator docetaxel which has demonstrated notable efficacy in preclinical models and warrants further exploration [13].

## 9. Conclusions

Supratentorial NSBMs have distinct clinical, pathological, and molecular characteristics from meningiomas located in the skull base and other locations. Although most supratentorial NSBMs are WHO grade 1, the high propensity to recur despite complete surgical resection make these challenging tumors, and further surgical intervention alone can result in potential patient morbidity and poor outcome [138]. The development and application of systemic therapies for refractory meningiomas is greatly needed.

The recent advances in molecular profiling of meningiomas may lead to biomarker designed clinical trials. Increased understanding of meningioma biology will hopefully translate into development of novel targeted therapy for these patients.

## Figures and Tables

**Figure 1 cancers-14-05887-f001:**
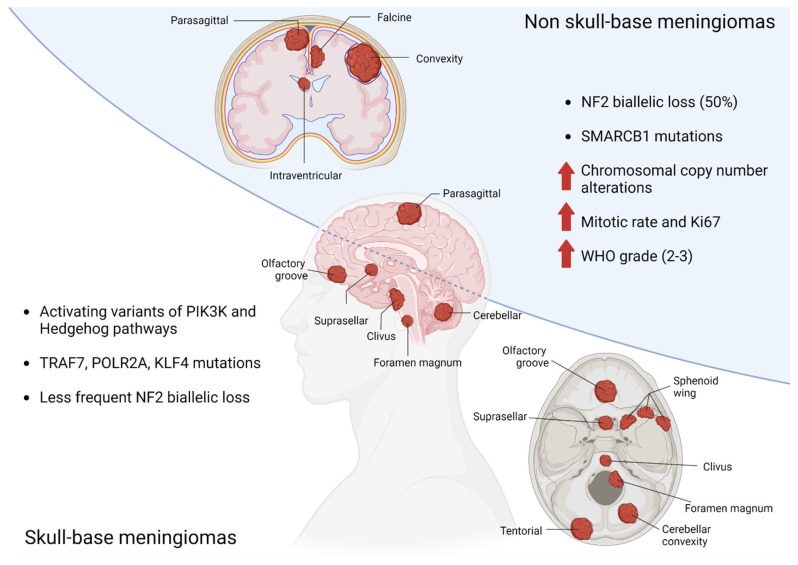
Genetic abnormalities related to anatomic location of meningiomas.

**Figure 2 cancers-14-05887-f002:**
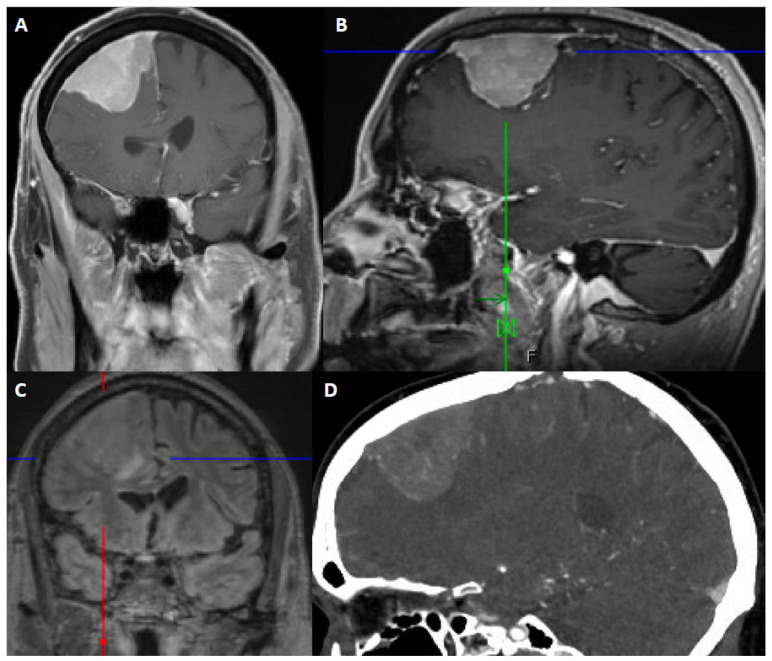
67-year-old male presented with seizure and fall. After resection, pathology showed atypical meningioma, WHO grade 2. Molecular markers: *TERT* promoter wild type, positive for loss of chromosome 1p, 6, 14, 22, Y, segmental loss at 9p (haploinsufficiency of *CDKN2A/B*). Patient underwent right middle meningeal artery embolization preoperatively. (**A**) Coronal contrast-enhanced T1-weighted image shows an avidly contrast-enhancing dural-based mass, dural thickening and enhancement, a slightly multilobular border, and moderate mass effect. (**B**) Sagittal contrast-enhanced Magnetization Prepared-RApid Gradient Echo (MPRAGE) shows a reactive dural tail. (**C**) Coronal 3D fluid attenuated inversion recovery (FLAIR) sequence shows associated vasogenic edema. (**D**) Sagittal CTA of the head demonstrates internal vascularity.

**Figure 3 cancers-14-05887-f003:**
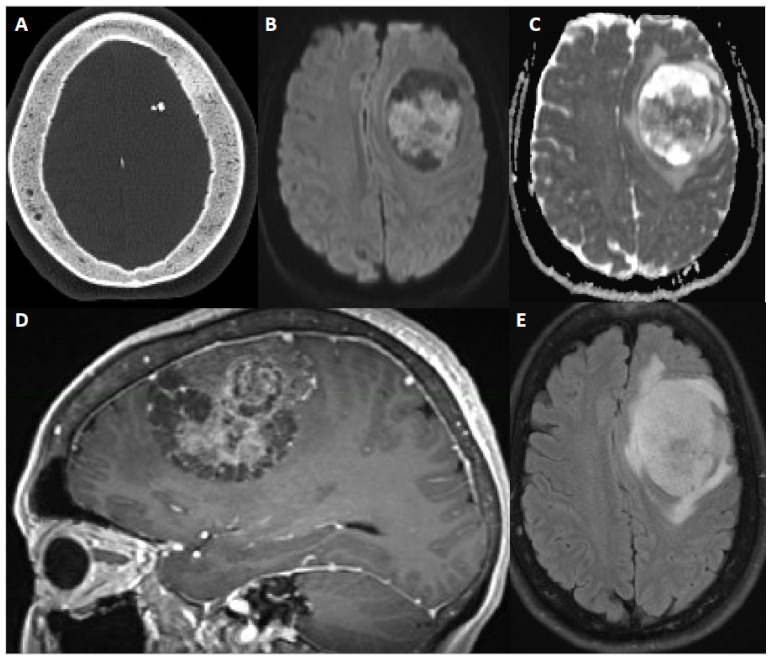
60-year-old female with new onset seizure, nausea, vomiting, and headache. After resection, pathology showed atypical meningioma, WHO grade 2. Molecular markers: *TERT* promoter wild type, positive for chromosome 8 and 22 loss (haploinsufficiency of *NF2*). (**A**) Axial CT head shows calcification within a mass and sclerosis of the overlying calvarium. (**B**,**C**) DWI and ADC map show restricted diffusion within the extra-axial mass. (**D**) Sagittal contrast-enhanced Magnetization Prepared-RApid Gradient Echo (MPRAGE) shows a heterogeneously enhancing multicystic dural-based mass with moderate mass effect. (**E**) Axial fluid attenuated inversion recovery (FLAIR) sequence shows moderate surrounding vasogenic edema.

**Table 1 cancers-14-05887-t001:** Clinical Series of Supratentorial Non-Skull Base Meningiomas Reported in the Literature.

Author	Year	Year Range of Data Presented	Number of Patients	Number of Patients with NSBMs	Patients with NSBMs (%)
Liang et al. [8]	2014	2009–2013	1239	629	50.7
Magill et al. [13]	2018	1985–2015	1113	431	39
Meling et al. [4]	2019	1990–2010	1148	586	51
Sun et al. [9]	2020	2012–2016	1107	535	48
Oya et al. [7]	2021	2001–2008	4081	2303	56.4

**Table 2 cancers-14-05887-t002:** Systemic Therapies for Meningioma.

Drug Class	Route of Delivery	Comment	Molecular Target
AKT inhibitor	Per Oral	Phase 2 trial	AKT1 mutation
Immune checkpoint inhibitor	Intravenous	Phase 2 trial	PD-L1, PD-L2, CTLA-4
Hedgehog inhibitor	Per Oral	Phase 2 trial	SMO mutation
PI3K inhibitor		Phase 2 trial	PI3K
Somatostatin analog	Per Oral	Phase 2 trial	Somatostatin receptors
Gemcitabine	Intraperitoneal	In vivo study	Cytidin
CDK inhibitor	Per Oral	Phase 2 trial	CDK mutation/NF2 loss
FAK inhibitor	Per Oral	Phase 2 trial	NF2 loss
Sunitinib	Per Oral	Phase 2 trial	Vascular endothelial growth factor receptor (VEGFR)
Bevacizumab	Intravenous	Phase 2	VEGFR
Docetaxel		In vivo and in vivo	G-protein coupled receptor (GPCR) signaling pathways

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
