# Peer review of "Clinical Management of Supratentorial Non-Skull Base Meningiomas"

_cancers, 2022, doi:10.3390/cancers14235887_

Round 1

Reviewer 1 Report

Thank you for the opportunity to review this review on the clinical management of meningiomas.

It is well structured and a good general overview. I feel there is little to criticize and it is more an editorial decision.

As a comment I would like to add, that I would change the wording, that meningiomas "arise from the fibrous coverings of the brain", because I find it misleading and would instead mention the arachnoidea and the arachnoid granulations.

I would also suggest to add a paragraph on currently discussed concepts of lymphatic system-like clearing in the meninges, perhaps also the influence of this on edema in meningiomas.

Also there needs to be more information on 850K Methylation essays. The possible role on clinical decisions, the current limitations. 

Author Response

We greatly appreciate the reviewer's favorable comments.

These have been addressed in our revised manuscript.  Specifically:

1) In the Pathophysiology section the wording has been changed to reflect the presumed arachnoid origin of meningiomas.

2) We added a paragraph (new 2nd paragraph of the Pathophysiology section) discussing meningeal lymphatics, an important topic we had previously omitted.  Additional citations of high-profile work in this field were added (Louveau, et al Nature and Nature Neuroscience.).  We all cited Choudhury, et al. Nature Genetics, which had been cited later in the Review.

3) Additional information regarding clinical application of DNA methylation profiling was added to the final paragraph of the Pathology and Molecular Diagnostics section.

Reviewer 2 Report

This manuscript elaborates on the latest insight into supratentorial non-skull base meningiomas in terms of pathophysiology, epidemiology, clinical manifestation, and management. A number of literature presented in a well-balanced way must be useful in daily practice for clinicians.

Although none of the radiotherapies has been proven to be effective for high-grade meningiomas so far, particle radiation therapies might be promising alternatives. El Shafie et al. reported a very good median PFS of 25.7 months in high-grade recurrent meningioma with carbon ion therapy. Miyatake et al. recently updated their outcomes of 44 patients with refractory high-grade meningiomas treated with reactor-based boron neutron capture therapy (BNCT). According to this study, the median PFS was 13.7 months and OS was 29.6 months. These particle therapies along with systemic therapies could improve outcomes of refractory meningiomas in the future. Thus, I would like to suggest referring to these studies.   

El Shafie RA, Czech M, Kessel KA, et al. Evaluation of particle radiotherapy for the re-irradiation of recurrent intracranial meningioma. Radiat Oncol. 2018;13(1):86

Takai S, Wanibuchi M, Kawabata S, et al. Reactor-based boron neutron capture therapy for 44 cases of recurrent and refractory high-grade meningiomas with long-term follow-up. Neuro Oncol. 2022;24(1):90–98.

Author Response

We appreciate the reviewers comment.

Additional discussion of particle based therapies has been added to the first paragraph of the Radiation section.  We have included the two references suggested by the reviewer, as well as one additional reference.

Reviewer 3 Report

Overall, this review is well-written and clearly organized. It comprehensively discusses the most up-to-date literature on supratentorial non-skull base meningiomas and details recent advances in genomics, radiomics and management. Just two very minor suggestions:

- clarify the use of WHO CNS grading toward the beginning of the paper, e.g., lines 57, 120

- consider rewriting the radiomics portion of the imaging section, as the discussion seems somewhat disjointed and cursory. Understandably, the field is nascent with many early and unvalidated claims. But as an example, the discussion of reference 56 (Shin et al, J Neurorad) starting at line 194 states that "the use of distinct [...] DWI parameters correlates with TERT promoter mutation status in grade 2 meningioma" and lists the statistically significant variables from the paper's univariate analysis (rather than the multivariate), which is somewhat confusing. The discussion of reference 72 (line 225) seems incomplete - what about DOTATATE PET predicts future growth? SUVmax or some other factor?

Author Response

We appreciate the positive feedback.

Additional clarification regarding WHO 2021 classification was added as requested.

We have also revised the Radiology section to improve clarity.
